# Genome-Wide Analysis of the SNARE Family in Cultivated Peanut (*Arachis hypogaea* L.) Reveals That Some Members Are Involved in Stress Responses

**DOI:** 10.3390/ijms24087103

**Published:** 2023-04-12

**Authors:** Chaoxia Lu, Zhenying Peng, Yiyang Liu, Guowei Li, Shubo Wan

**Affiliations:** Shandong Provincial Key Laboratory of Crop Genetic Improvement, Ecology and Physiology, Shandong Academy of Agricultural Sciences, Jinan 250100, China

**Keywords:** peanut, SNARE, gene family, vesicular transport, stress

## Abstract

The superfamily of soluble N-ethylmaleimide-sensitive factor attachment protein receptor (SNARE) proteins mediates membrane fusion during vesicular transport between endosomes and the plasma membrane in eukaryotic cells, playing a vital role in plant development and responses to biotic and abiotic stresses. Peanut (*Arachis hypogaea* L.) is a major oilseed crop worldwide that produces pods below ground, which is rare in flowering plants. To date, however, there has been no systematic study of SNARE family proteins in peanut. In this study, we identified 129 putative SNARE genes from cultivated peanut (*A. hypogaea*) and 127 from wild peanut (63 from *Arachis duranensis*, 64 from *Arachis ipaensis*). We sorted the encoded proteins into five subgroups (Qa-, Qb-, Qc-, Qb+c- and R-SNARE) based on their phylogenetic relationships with *Arabidopsis* SNAREs. The genes were unevenly distributed on all 20 chromosomes, exhibiting a high rate of homolog retention from their two ancestors. We identified cis-acting elements associated with development, biotic and abiotic stresses in the promoters of peanut SNARE genes. Transcriptomic data showed that expression of SNARE genes is tissue-specific and stress inducible. We hypothesize that *AhVTI13b* plays an important role in the storage of lipid proteins, while *AhSYP122a*, *AhSNAP33a* and *AhVAMP721a* might play an important role in development and stress responses. Furthermore, we showed that three AhSNARE genes (*AhSYP122a*, *AhSNAP33a* and *AhVAMP721*) enhance cold and NaCl tolerance in yeast (*Saccharomyces cerevisiae*), especially *AhSNAP33a*. This systematic study provides valuable information about the functional characteristics of AhSNARE genes in the development and regulation of abiotic stress responses in peanut.

## 1. Introduction

Vesicular trafficking is a fundamental system maintaining cellular functions in all eukaryotic cells and is essential for development and adaptation. Communication routes among distinct organelles require vesicular trafficking, through which proteins and soluble cargo are sorted to their correct compartments [1]. The immobile and multicellular nature of plants requires the constant monitoring of changes in the environment and rapid reprogramming of metabolic and gene expression profiles adapted to adverse conditions, such as soil salinity, drought, extreme temperatures, nutrient imbalances and the presence of toxic metals. Cellular responses to various environmental stresses rely on the regulation of vesicular trafficking to ensure proper localization of proteins specialized in sensing stress stimuli and effecting the appropriate response. Membrane fusion, the last step of vesicular trafficking, is promoted by complex formation of soluble N-ethylmaleimide-sensitive factor attachment receptor proteins (SNAREs) [2]. SNAREs are located in different subcellular organelles and take part in a series of fundamental processes, for instance, cytokinesis, cytoskeleton organization, symbiosis, growth, and biotic and abiotic stress responses [3,4,5,6]. For instance, the SYP4 (SYNTAXIN OF PLANTS 4) and SYP6 groups are located in TGN and positively affect tolerance to salinity and osmotic stresses [7,8]. The identified immune protein of the powdery mildew RPW8.2 is transported by AtVAMP721/722 (vesicle-associated membrane protein 721/722) vesicles [9].

SNAREs are well conserved within eukaryotes, with genomic studies demonstrating that plants have considerably more *SNARE* genes than other eukaryotes. The SNARE family consists of 21–25 and 35–36 family members in yeast (*Saccharomyces cerevisiae*) and humans (*Homo sapiens*), respectively, while there are 36 SNAREs in the liverwort *Marchantia polymorpha*, 64 in Arabidopsis (*Arabidopsis thaliana*) and 60 in rice (*Oryza sativa*) [10,11,12]. SNARE proteins contain 100–400 amino acids identified by a coiled-coil SNARE domain and can be divided into R-SNARE and Q-SNARE proteins according to the conserved amino acid residues (Arg or Glu) in their central structure. Q-SNAREs are often located on the cytoplasmic membrane and are also known as t-SNAREs; R-SNAREs are located on vesicle membranes and are also called v-SNAREs [3,13,14]. Q-SNAREs can be further divided into four subgroups: Qa-, Qb-, Qc- and Qb+c-SNAREs [15]. Most SNAREs connect with lipid bilayers via a C-terminal transmembrane domain, but some exceptions attach to membranes via a lipid anchor [16,17]. When vesicles fuse, four different SNAREs (Qa-Qb-Qc-R or Qa-Qb+c-R) interact to form a stable complex called the SNARE complex, facilitating fusion of the phospholipid bilayer [18].

The plant vesicular trafficking system is considerably complex. It is characterized by expanded protein families that increase the specificity of vesicular trafficking as well as by plant-specific features that have evolved to perform specific functions in the plant lifestyle [5,11,19]. Novel Plant-specific SNARE (NPSN) and SYP7 Qc-SNARE proteins are specific to plants [20]. These additional proteins may allow plants to mount a coordinated response to the environment and confer special properties to some plants. Many crops, including wheat (*Triticum aestivum* L.), maize (*Zea mays* L.), rice, beans (*Phaseolus vulgaris* L.) and peanuts, are major sources of protein for humans and livestock. These crops accumulate storage proteins in subcellular organelles such as the endoplasmic reticulum and vacuoles. Therefore, the accumulation, transport and localization of many storage proteins are also dependent on the vesicular transport system. There is potential to improve yield, nutrition and stress tolerance by improving the vesicular transport system [21].

Cultivated peanut (*Arachis hypogaea* L.) originates from South America. It is a major oil crop grown in semi-arid and arid hilly areas with high tolerance to drought and infertile conditions [22]; yield per unit is higher than that of other oil crops [23]. Peanut growth is very sensitive to environmental stress, which seriously affects yield and quality. To date, the molecular characterization of SNAREs in peanut has identified few genes [24], and a systematic study of the peanut SNARE family has not been performed. Cultivated peanut is an allotetraploid (AABB), and molecular evidence indicates that *Arachis duranensis* Krapov. & W.C Greg. and *Arachis ipaensis* Krapov. & W.C Greg. are the donors of the A and B genomes, respectively [25]. Whole-genome sequences and annotation of *A. hypogaea* ‘Tifrunner’ [22,26] now make it possible to identify all the putative *SNARE* genes in cultivated peanut and to analyze their potential functions in different tissues and under abiotic stress conditions.

To promote future research on the SNARE family in this species, we identified all *SNARE* genes in the genomes of peanut (*A. hypogaea*, *A. duranensis* and *A. ipaensis*). We also analyzed the structure, chromosomal location and conserved domains of *SNARE* genes using bioinformatics methods. In addition, we analyzed the transcriptional profiles of *SNARE* genes at different developmental stages and performed a comparative analysis of expression levels after different treatments to characterize the mechanisms governing vesicular transport in response to different abiotic stresses and identify candidate genes for peanut improvement. Importantly, we verified the function of the putative SNARE genes’ response to salt and chilling stress in *Saccharomyces cerevisiae.*

## 2. Results

### 2.1. Identification and Classification of SNAREs in Peanut

We identified 129 SNARE proteins from *A. hypogaea* L. after screening the peanut protein database with the conserved domain PF05739 (Appendix A). We also identified SNAREs in the two diploid ancestors of cultivated peanut: 63 in *A. duranensis* and 64 in *A. ipaensis* (Appendix A). All candidate SNAREs contained the conserved domain PF05739 in peanut, similar to Arabidopsis, but also contained other conserved domains such as syntaxin, sec20, use1, longin and synaptobrevin (Appendix A).

Among the SNAREs identified in *A. hypogaea*, 34 were syntaxins—the prototype family of SNARE proteins—including Syntaxin1, Syntaxin2, Syntaxin5, Syntaxin18 and SynN. Syntaxin18 and Syntaxin5 are found in the SNARE complex of the endoplasmic reticulum (ER) and play an important role in transport between the intermediate compartment of the ER and the *cis*-Golgi vesicle [27]. The N-terminal region of Syntaxin18 is especially important for the formation of ER aggregates [28]; Syntaxin5 (Syn5) participates in the assembly of transitional ER and the Golgi, lipid droplet fusion and cytokinesis [29]. Another conserved family with 32 members identified in cultivated peanut was Synaptobrevin or R-SNAREs, which localize at vesicle membranes [7]. Some of these R-SNAREs contained a longin domain, which is a motif of proteins transporting vesicles from the ER to the plasma membrane via the Golgi apparatus. SNARE-conserved domain GS27 was identified in *AhMEMB1* (a Qb-SNARE membrin), a member of the Qb subfamily of SNARE proteins involved in regulating the early secretory pathway of eukaryotic cells [30].

As shown in Appendix A, the length of *SNARE* genes ranged from 516 bp (*AhVAMP713b*) to 13,843 bp (*AhTYN12d*), encoding proteins with 124 (*AhBET11a*) to 1883 (*AhTYN11b*) amino acids in length; these proteins were mostly between 124 (*AhMEMB11b*) and 426 (*AhSYP111b*) amino acids, except for those of the AhTYN subfamily. The predicted molecular weights (MW) of the proteins were mostly between 12.8 kDa (*AhSFT12c*) and 48.3 kDa (*AhNPSN12b*); however, those belonging to the AhTYN subfamily were far beyond this range at 105.8~125.4 kDa. Predicted PIs (isoelectric points) varied from 4.75 (*AhSYP23c*) to 9.99 (*AhSYP81b*), with an average of 7.46, among which 69 AhSNAREs had pI > 7 and 60 AhSNAREs had pI < 7. Subcellular localization predictions suggested that peanut SNAREs are predominantly located at the plasma membrane, ER/Golgi and trans-Golgi network (TGN)/vacuole/endosome (Appendix A).

### 2.2. Peanut SNARE Genes Exhibit a High Rate of Homoeolog Retention

To gain insight into the relationship and classification of *AhSNARE* genes, we reconstructed a maximum-likelihood phylogenetic tree with 129 AhSNARE proteins from cultivated peanut, 63 AdSNAREs from *A. duranensis*, 64 AiSNAREs from *A. ipaensis* and 64 AtSNAREs from Arabidopsis. The cultivated peanut genome retained all major SNARE gene subfamilies: BET1, GOS1, MEMB1, NPSN1, SEC20, SEC22, SFT, SNAP, SYP1(SYP11, SYP12, SYP13), SYP4, SYP5, SYP6, SYP7, SYP8, TYN1, USE1, VAMP, VTI and YKT6 (Figure 1 and Appendix A). The above phylogeny roughly followed species phylogeny; *A. hypogaea* genes displayed a sister–group relationship with Arabidopsis genes in many subclades (Figure 1). Almost all *A. hypogaea* genes were closely related to their two homologous ancestral sequences; sequence similarity was more than 85%. Many subclades showed the expected 2:1 ratio of cultivated peanut-to-Arabidopsis genes, the sum of two ancestral subfamily members (SEC 20, SEC 22, SFT, SNAP); however, some ratios were lower than expected (e.g., NPSN) (Figure 2). These results suggested that expansion of *SNARE* homologs in *A. hypogaea* mainly resulted from heterologous hybridization. The *AhBET* Qc-SNARE subfamily expanded and experienced duplications in cultivated peanut after hybridization. The increased number of BET subfamily members in peanut suggests the possibility of novel functions for one or more family members [21].

### 2.3. Conserved Domain and Gene Structure Analysis

Analysis of conserved motifs and gene structures of each *AhSNARE* using TBtools revealed that members of the same subfamily share similar motifs (Figure 3B). Motifs 1 and 8 corresponded to the longin and synaptobrevin domains, respectively, which were present in R-SNAREs; Motif 20 was the syntaxin domain, which was present in Qa-SNAREs; Motifs 23–24 were V-SNARE-C, which were present in Qb-SNAREs; Motif 10 was Qc-SNARE. SYP2 possessed Motifs 7, 9 and 20; SNAP possessed Motifs 10 and 22.

A bioinformatics analysis of gene structure can be useful for recognizing patterns of intron acquisition and loss during evolution of a family whose structure and function are well known. A summary of the exon–intron structure of each *AhSNARE* gene is shown in Figure 3C. The numbers and lengths of introns showed patterns reflecting the organization of the AhSNARE family in the phylogenetic tree (Figure 3A). However, genes from different groups varied widely in size and intron arrangement. In the SYP1 clade, genes of the AhSYP11 and AhSYP12 subfamily had only one or no intron, while there were 12 introns in genes of the AhSYP13 subfamily. The SYP1 clade is reported to split into two groups, SYP12 and SYP13, among embryophytes. A common feature of the SYP13 is that all genes have multiple introns, almost all of which occur at similar positions within the coding region [31]. Furthermore, there were 23 introns in genes of the AhTYN1 subclade, and the structures of these genes showed many variations.

### 2.4. Chromosomal Mapping and Duplication Analysis of Peanut SNARE Genes

To determine the distribution of *AhSNARE* genes on chromosomes, we drew a physical map using TBtools onto which we mapped all 129 *AhSNARE* genes. These genes were distributed unequally on the 20 chromosomes (Figure 4), with 80% of *AhSNAREs* located in the distal regions of chromosomes. The fewest *SNARE* genes were on chromosomes 6 and 13, which only had three genes each, while chromosomes 1 and 20 harbored the highest number of *SNARE* genes (10 genes each). Homoeologous chromosomes from the A and B genomes, such as chromosomes 4 and 14 (6 genes), 7 and 17 (8 genes), and 9 and 19 (5 genes), typically had similar numbers of *SNARE* genes.

To confirm duplication events between genes, were performed collinearity analysis between *A. hypogaea* and *A. duranensis*, *A. hypogaea* and *A. ipaensis* (Figure 5A), and between the A and B genomes of *A. hypogaea* (Figure 5B). We observed high rates of gene retention: 91.5% of *A. hypogaea* genes were homologous to those of the proposed ancestors. The Ka/Ks ratios for each pair of homologous genes were generally less than 1 (Appendix A), except those for *AdBET11*-*AhBET11b* (Ka/Ks = 2.042), *AdVAMP721*-*AhVAMP721b* (Ka/Ks = 1.161), *AiVAMP711*-*AhVAMP711b* (Ka/Ks = 2.042) and *AiSFT12*-*AhSFT12a* (Ka/Ks = 1.664). We identified 10 dispersed duplicated genes in *A. hypogaea* (*AhBET11d*, *AhSYP121b*, *AhVAMP714b*, *AhBET12c*, *AhBET11c*, *AhBET12b*, *AhNPSN13b*, *AhMEMB11b*, *AhSYP112b* and *AhMEMB12c*). According to the phylogenetic tree, three pairs of dispersed duplicated genes belonged to the Qb-SNARE subfamily, four belonged to the Qc-SNARE subfamily, one belonged to the R-SNARE subfamily and two belonged to the Qa-SNARE subfamily.

### 2.5. Analysis of Cis-Acting Elements in Promoters of AhSNARE Genes

To gain insight into the *cis*-acting elements of *AhSNAREs*, we retrieved sequences 2000 bp upstream of the translation start position of all *AhSNAREs* from the peanut genome database. We identified many response-related elements within these regions (Appendix A), such as MYB recognition, TGAACG-motif, ARE (Antioxidant response element), ABRE (ABA-response element), STRE (Hot responsive element), TC-rich repeats and LTR (Low temperature response element). These *cis*-acting regulatory elements are associated with plant biotic and abiotic stress responses. The TGAACG-motif and ABRE are *cis*-acting regulatory elements involved in responsiveness to methyl jasmonate and abscisic acid (ABA), respectively. We also identified some development-related elements, such as CAT-box and the GCN4_motif. These results suggest that *AhSNAREs* are involved in responses to environmental stress, as well as growth and development of peanut.

### 2.6. Differential Expression Profiles of SNARE Genes in Different Tissues and at Different Developmental Stages of Peanut Pod

Peanut plants bear fruit underground; groundnut seeds contain 22% to 30% protein and 35% to 60% oil. To understand the functions of *AhSNARE* genes in peanut development and growth, we analyzed their expression profiles at different developmental stages of peanut pods and in different tissues. We obtained transcriptome data for 118 out of 129 SNARE genes, with 90% of these genes being expressed in at least one tissue or at one developmental stage, with a wide expression range from 1 to 185 FPKM. The remaining 10% of *SNARE* genes displayed very low expression levels below 1 FPKM (Figure 6) and were considered not expressed. We also analyzed the expression levels of homoeologous *SNARE* gene pairs from the A and B subgenomes. Unlike results reported from the peanut genome [26], more homoeologous gene pairs exhibited expression bias towards the A subgenome than towards the B subgenome.

We used phylogenetic classification to draw five heatmaps of the 118 expressed *AhSNARE* genes to evaluate their transcript abundance in nine tissues and at 12 pod developmental stages (Figure 6). The expression profile of each *SNARE* gene was different, although certain genes in different groups had similar expression patterns, which may provide clues as to their interactions. Several genes in each group were highly expressed in all tissues and at all developmental stages analyzed (e.g., *AhSYP32* and *AhSYP21* in Qa-SNARE; *AhUSE1a*, *AhMEMB11b* and *AhVTI11a* in Qb-SNARE; *AhSYP61b*, *AhSYP52b* and *AhSYP71a/b* in Qc-SNARE; *AhVAMP726*, *AhVAMP 724a* and *AhVAMP721* in R-SNARE), suggesting that these genes are likely to play a regulatory role during the entire growth period of peanut. Several genes showed higher expression in roots, nodules and pericarp (*AhSYP122* in Qa-SNARE; *AhSFT12* in Qc-SNARE; *AhSNAP29b* and *AhSNAP33a* in Qa+c-SNARE). Qa-SNARE *AhSYP131*, *Qc-SNARE AhSYP73*, Qb-SNARE *AhNPSN11b/d*, Qb+c-SNARE SNAP30a/b and V-SNARE *AhVAMP713a/b* were specifically expressed in pistils. *AhSYP22a/b*, *AhSYP23a/b*, *AhMEMB11b* and *AhYKT62a* reached their highest expression levels during the latest stage of peanut pod development. As the seed matured, the expression level of *AhVTI13a/b*, expressed in the TGN/endosome/vacuole, rose markedly. However, lower expression of *AhSFT12a/b* and *AhSYP121a/b* was observed during the later stage of peanut seed development, with higher expression during the early stage of peanut seed development. The gene encoding Qa-SNARE AhSYP122a, which is located at the plasma membrane, showed high expression levels in Pattee 5 pericarp and Pattee 6 pericarp stages but showed very low transcript levels in peanut kernel.

### 2.7. Expression Patterns of AhSNARE Genes under Salt and Low-Temperature Stresses

*SNARE* genes play a vital role in abiotic and biotic stresses [3,32,33,34,35]. To understand the roles of *SNARE* genes in peanut, we summarized their expression patterns by analyzing transcriptomic data of peanut seedlings exposed to diverse stresses (salt, drought, high/low temperature, ABA, sodium nitroprusside, H_2_O_2_ and methyl jasmonate) for 24 h (Appendix A). *SNARE* genes in roots and leaves were regulated differentially by different stresses; most homologous gene pairs had similar expression patterns, and the expression difference was obvious at low temperature. Expression levels of *AhSYP122a/b* in the Qa-SNARE subfamily, *AhSFT12a/b* in Qc-SNARE, *SNAP29a/b* in Qb+c-SNARE and *AhVAMP725/726* in R-SNARE were significantly higher after low-temperature treatment for 24 h. Few *SNARE* genes showed a significant difference in their expression levels under salt stress compared to control conditions (Appendix A). However, salt and chilling stress limit the productivity of peanut, damage photosystem complexes and change the fatty acid composition of peanut leaves [36,37].

We selected members of each subfamily to validate peanut *SNARE* expression patterns after salt and chilling stress. As shown in Figure 7, we examined 10 genes from different subfamilies—*AhSYP122a* (Qa), *AhSYP132a* (Qa), *AhSYP22a* (Qa), *AhNPSN12b* (Qb), *GOS12a* (Qb), *AhSNAP33a* (Qb+c), *AhSFT12a* (Qc), *BET12b* (Qc), *VAMP725* (R) and *VAMP721a* (R)—at five time points (0, 3, 12, 18 and 24 h) in leaves and roots of FH1. These *AhSNARE* genes showed different expression profiles in response to low-temperature and salt treatments. Under low-temperature treatment, *AhSFT12a* and *AhSYP122a* were highly expressed in both roots and leaves, with expression values reaching 15–20 times those of untreated samples at 18 h. Expression levels of *AhSYP132a* and *AhSYP22a* displayed an early (3 h and 6 h) upregulation followed by downregulation in leaves under low-temperature treatment; however, there was little difference in their expression in roots. The expression levels of *AhGOS12a*, *AhSYP122a*, *AhVAMP725* and *AhVAMP721a* reached a maximum value at 18 h in roots under low-temperature treatment before dropping. *AhBET12b* expression was suppressed in leaves under low-temperature treatment but was elevated in roots. After salt treatment, the expression level of *AhSYP122a*, *AhNPSN12b*, *AhGOS12a*, *AhSNAP33a*, *AhVAMP721a* and *AhVAMP725* was more than five times that of the control group in roots. *AhSYP122a*, *AhNPSN12b* and *AhGOS12a* transcript levels responded quickly to salt treatment, with expression levels reaching a maximum at 3 h before declining. The expression of *AhSYP132a* and *AhSYP22a* was not significantly different in roots throughout salt treatment and reached a maximum at 12 h after salt treatment. As shown in Figure 7, *AhSYP122a*, *AhSNAP33a* and *AhVAMP721a* expression was more than ten times that before treatment under both salt and low-temperature stress, and we speculated that these genes play a key role in abiotic stress responses.

### 2.8. AhSYP122a, AhSNAP33a and AhVAMP721a Enhance Saccharomyces Cerevisiae Growth under Cold and NaCl Stress

To study the potential function of AhSNAREs under stress conditions, we cloned the full-length coding sequences of three genes: *AhSYP122a*, *AhSNAP33a* and *AhVAMP721a.* We investigated the function of these three AhSNAREs under cold and NaCl stress in budding yeast (*Saccharomyces cerevisiae*). Accordingly, we individually transformed yeast strain BY4741 with the plasmids pESC-*AhSYP122a*, pESC-*AhVAMP721a*, pESC-*AhSNAP33a*, pESC-*AhSYP122a/AhSNAP33a* or pESC-*AhVAMP721a/AhSNAP33a* allowing the expression of each selected gene. We then exposed the resulting positive yeast colonies to cold stress at −20 °C for 1 h, or to growth on medium containing 0.5 M NaCl or 0.8 M NaCl for 5 h. As illustrated in Figure 8, there was no substantial difference between BY4741 expressing *AhSNARE* genes and BY4741 harboring the empty vector pESC under normal conditions. However, BY4741 containing pESC-*AhSYP122a*, pESC-*AhVAMP721a* or pESC-*AhSNAP33a*, particularly the latter, had a higher viability than yeast containing empty vector pESC after NaCl or freezing (−20 °C) stress. In addition, BY4741 heterologously expressing two *SNARE* genes (pESC-*AhSYP122a/AhSNAP33a* or pESC-*AhVAMP722a/AhSNAP33a*) had slightly greater viability under NaCl or freezing (−20 °C) than yeast cells expressing only one *SNARE* gene. This result suggests that heterologous expression of *AhSYP122a*, *AhSNAP33a* or *AhVAMP721a* enhanced tolerance to abiotic stress in yeast.

## 3. Discussion

SNAREs are known to perform crucially important roles in vesicular trafficking and participate in multiple growth and stress responses. Therefore, they are expected to be a target for crop breeding and improvement. The SNARE multigene family has been studied and analyzed in several plant species, for instance, Arabidopsis, rice [31], tomato (*Solanum lycopersicum*) [13], olive rape (*Brassica napus*) [38] and wheat (*Triticum aestivum*) [39]. Peanut is a special leguminous crop with subterranean pods and high oil and protein contents. Therefore, it is important to identify all putative *SNARE* genes in cultivated peanut and analyze their potential functions.

Here, we identified 129 *SNARE* genes from *A. hypogaea*, 63 from *A. duranensis* and 64 from *A. ipaensis*, which were assigned to 21 conserved subfamilies. Studies on the *SNARE* family in other species have indicated that the number of *SNAREs* may be relevant to plant polyploidy. For instance, there are 63 *SNAREs* in tomato (AB) [28] and 173 in wheat (AABBDD) [39]. In cultivated peanut (*A. hypogaea*), 90.0% of peanut *SNARE*s can be assigned to 1:1 homoeologous genes from *A. duranensis* and *A. ipaensis* [25]. This is a considerably high homoeologous retention rate. The high numbers of *AhSNARE* genes might reflect the allotetraploid genome of peanut. Five out of twenty-one subfamilies did not appear as pairage in our study (Appendix A and Figure 2). For instance, the number of *AhNPSN1* subfamily members was lower in *A. hypogaea* than in the probable progenitors combined, suggesting loss of these genes during long-term evolution or insufficient sequencing depth. By contrast, the number of *AhBET* subfamily members in *A. hypogaea* was three more than the sum from the two ancestral genomes. *AtBET11* (also named *Bs14b*) and *AtBET12* (also named *Bs14a*) localize to the Golgi membrane and affect protein ER–Golgi anterograde transport. Previous studies have shown that *AtBET11* and *AtBET12* are necessary for the growth and development of pollen tubes, with reduced seed fertilization, defective embryo development, and reduced pollen tube length and secondary tube formation observed in the Arabidopsis *bet11 bet12* double mutant [40]. SNARE proteins act in multi-protein complexes; R-SNARE, Qa-, Qb- and Qc-SNARE self-assemble into a complex of four coiled-coil helices to complete the fusion of vesicles and target membranes [41]. Changes in gene dosage may lead to changes in the possible combinations of proteins in these complexes, which may result in phenotypic or functional effects [42]. A large number of *AhBET* genes may be related to the special flower development of peanut. Through gene duplication and collinearity analysis, we determined that 91.5% of *SNARE* genes are homologous to their ancestral genes. Once more, this result demonstrates a high rate of homoeolog retention after the hybridization of the A and B genomes (Figure 5) and the biological importance of the *SNARE* gene family in general.

We identified a large number of *cis*-acting elements related to various stresses, phytohormones and developmental processes in the *AhSNARE* promoters of peanut (Appendix A). Most of the SNARE proteins studied participate in responses to abiotic and biotic stress [32,43,44] and the regulation of development [5,29,45]. The peanut has one of the most interesting growth habits: it produces flowers aerially and bears fruit underground. The ovary is not surrounded by petals but is located instead at the base of the hypanthium. After fertilization, the embryo undergoes a few rounds of cell division and then becomes dormant. The meristem of the ovary then begins to elongate and forms a “peg” structure, with the ovary just behind the lignified top. Our expression data suggested that *AhSYP131b*, *AhSYP72*, *AhNPSN11a/b*, *AhVAMP713* and *AhVAMP723* were specifically expressed in pistils. It is possible that SNARE genes play a key role in the special reproductive mode of peanuts. Interestingly, *AhSYP131b*, *AhNPSN11a/b* and *AhVAMP723* are also related to pollen tube development [46,47]. Once the tip is a few centimeters below the soil surface, the pod begins to develop. Most of the cell division occurs in the distal region 1–3 mm from the tip [48]. In our study, *AhSYP11* showed the highest expression during the development of the peg (Figure 6). AtSYP111 (also named KNOLLE), the first SNARE family member discovered, and its interacting protein KEULE (Sec1/Munc18), play key roles in cell-plate formation and their encoding genes are specifically expressed during cell division [49]. In the *knolle* and *keule* mutants, many unfused vesicles aggregate at the cell plate [49,50,51]. Several studies showed that indole-3-acetic acid (IAA) plays the most important role in the development of gynophore and peanut fruits [48,52]. Immunogold electron microscopy (EM) study of *Arabidopsis* root apices displayed that there was a large percentage of IAA clustered within vesicles and membranous compartments [53]. Furthermore, the expression of *AhSYP122a* and *AhVAMP725* was higher in pericarp. Fibrous materials were deposited in the pericarp throughout maturation. Cellulose, synthesized by cellulose synthase complexes (CSCs), were transported to the plasma membrane by SYP61 labeled vesicles; the SYP121 complex was also contained in these vesicles.

Peanut is rich in protein and lipid, and its lipid content reaches its maximum in the later stages of development. The expression data in this study also showed that *AhVTI13b* in the Qb-SNARE subfamily was expressed at high levels during the late stage of seed development and might play a vital role in peanut lipoprotein storage. Previous studies demonstrated that *AtVTI12,* a homolog of *AhVTI13*, mediates trafficking to storage vacuoles [21]. SNARE proteins of different classes interact with each other to promote vesicle fusion [54]. Co-expression of genes from different groups provides a basis for analyzing the potential interaction of their encoding proteins, for example, *AhSYP122*, *AhSYP132*, *AhSNAP33* and *AhVAMP725* show higher expression in roots, nodules and pericarp than in other tissues. A recent study showed that the infection thread (IT) is not elongated in roots of birdsfoot trefoil (*Lotus japonicus*) knocked down for *LjSYP132* by RNA interference (*LjSYP132a-*RNAi, *LjSYP132b-*RNAi) [55]. Nitrogen fixation activity and the number of bacteroides severely decreased in nodules formed on *LjVAMP72a/72b-*RNAi knockdown roots, and arbuscular mycorrhization (AM) was also curtailed. Kim S. et al. found that the SYP132–VAMP721/722 interaction is involved in immunity to *Pseudomonas syringae* in tomato [56]. In Arabidopsis, most vesicular trafficking to the plasma membrane is driven by SYP121/122 through their assembly with the R-SNAREs VAMP721/722 and the Qb+c-SNARE SNAP33 [57]. Wang and Zhang found that *SYP121* and *SYP122* are hub genes that play a key role in plant growth-to-defense transition [58].

Plant responses to abiotic stress rely on the regulation of vesicular trafficking to ensure that proteins specialized for sensing stimuli and responding to stress are correctly localized. Drakakaki et al. (2012) isolated the SYP61 labeled compartment and carried out proteomic analysis of its contents; the SYP121-complex, cellulose synthases, and PIP 2;1 were identified [59]. Previous studies have shown that plant cells produce a large amount of reactive oxygen species under high salt environment, and the inhibition of VAMP7 gene expression leads to the failure of vesicles containing H_2_O_2_ fused with tonoplast and the formation of giant vesicles containing H_2_O_2_ in the cytoplasm. Under salt stress, the vesicles are very active, and Na^+^ accumulates not only in the main vacuole but also in the small vesicles around the vacuole [60]. Lu et al. (2020) also found that vesicular transport is involved in the secretion of salt in the leaves of *Limonium bicolor* [32]. Peanut is a thermophilic, drought-tolerant plant that is relatively sensitive to salt and low temperature. We studied the transcript levels of *AhSNARE* genes under salt and low-temperature conditions, which provided clues for exploring their function. Improper low-temperature stress can increase cell membrane permeability and membrane lipid peroxidation. Low temperature has a serious effect on peanut seedlings, with many differentially expressed genes identified at low temperature; however, there was no distinct difference in expression under salt treatment (Appendix A). RT-qPCR analysis over a 24 h time course revealed that many *AhSNARE* genes were highly expressed at 3–6 h in response to stress and their expression levels decreased at 24 h into salt treatment (Figure 7); expression levels of *AhSYP122a*, *AhSNAP33a*, *AhVAMP725* and *AhVAMP722a* were different after salt and low-temperature treatments. Most SNARE proteins reported to date participate in responses to abiotic and biotic stress [3]. VAMP71 proteins play a vital role in the localization of reactive oxygen species, and suppression of *AtVAMP7C* expression increases tolerance to salt stress [34,61]. SYP121/122 are redundant in their function, facilitating the majority of secretory trafficking to the plasma membrane but mediating different secretory cargoes [62]. SYP121/122 are involved in responses to pathogen invasion and abiotic stress, and the AtSYP121-AtSNAP33-AtVAMP721 SNARE complex participates in transport of K^+^ channels to the plasma membrane in Arabidopsis [57]. Furthermore, AtSYP121 together with AtSYP61 regulates the water potential of cells by coordinating the transport of aquaporin AtPIP2;7 (plasma membrane intrinsic protein 2;7) [59,63]. Heterologous expression of *AhSYP122a*, *AhVAMP721a* and *AhSNAP33a* increased cold and salt tolerance in yeast (Figure 8). However, heterologously expressing two SNARE genes (pESC-AhSYP122a/AhSNAP33a or pESC-AhVAMP722a/AhSNAP33a) had slightly greater viability than that in one SNARE gene. This is due to the snares’ needs to form a complex to function [7]. Salinas-Cornejo et al. (2021) reported that tomato overexpressing *SlSNAP33.2* displays more tolerance to salt stress than control tomato plants [64]. In addition, when wild soybean (*Glycine soja*) *SNAP33* was heterologously expressed in Arabidopsis, the transgenic plants showed improved salt and drought tolerance [65]. In Arabidopsis, AtSYP121 and AtVAMP721/722 are necessary for both biotic and abiotic stress responses [66]. AhSNAREs are engaged in multiple stress responses. Therefore, we believe that *AhSNARE* is a multifunctional gene family that plays vital roles in peanut growth and development as well as in defense responses to various environmental stresses.

## 4. Materials and Methods

### 4.1. Identification of the SNARE Family in Peanut

Genome sequences, functional annotations and coding sequence (CDS) predictions for *Arachis hypogaea*, *Arachis duranensis* and *Arachis ipaensis* were downloaded from Peanutbase (https://peanutbase.org/peanut_genome, accessed on 24 May 2022). SNARE family proteins were identified in the peanut protein database using HMMER 3.0 software with the HMM (Hidden Markov Model chain) file of the domain (PF05739) downloaded from the Pfam database. Incomplete and redundant amino acid sequences were removed. Splice variants were deleted, and only the longest variant was retained for further analysis. SNARE protein sequences from Arabidopsis were obtained from the TAIR database (https://www.arabidopsis.org, accessed on 22 May 2022). Additionally, each protein was manually validated using BLASTp (https://blast.ncbi.nlm.nih.gov, accessed on 24 May 2022), the online CD-search tool (http://www.ncbi.nlm.nih.gov/Structure/cdd/wrpsb.cgi, accessed on 28 May 2022) and the SMART tool (embl-heidelberg.de, accessed on 28 May 2022). Candidate SNAREs were named according to their most probable homolog in Arabidopsis, preceded by an abbreviation for the species name (*Ah for A. hypogaea*, *Ad for A. duranensis* and *Ai for A. ipaensis*); genes belonging to one subfamily but found in different genomes were numbered consecutively. The subcellular localization of SNAREs was predicted using Plant-mPLoc (http://www.csbio.sjtu.edu.cn/bioinf/plant-multi/, accessed on 2 June 2022) and ProtComp (http://linux1.softberry.com/berry.phtml, accessed on 2 June 2022).

### 4.2. Phylogenetic Analysis, Gene Structure and Conserved Motif Analyses of the SNARE Family

Alignment of full-length SNARE protein sequences from *A. hypogaea*, *A. ipaensis*, *A. duranensis* and Arabidopsis was performed using the MUSCLE program in the MEGA 7 software [67]. On the basis of this alignment, an unrooted phylogenetic tree was reconstructed using the maximum-likelihood method and the nearest-neighbor-interchange model. The resulting tree file was visualized using Evoview version 2.0 (https://www.evolgenius.info/evolview-v2, accessed on 24 July 2022). Conserved motifs in SNARE proteins were analyzed using an online tool (http://meme-suite.org/, accessed on 28 May 2022), allowing any number of repetitions with an optimum motif width of 6–50 residues and up to 30 motifs. Gene structure information was downloaded from the peanut database and assessed using an online tool (http://gsds.cbi.pku.edu.cn, accessed on 22 May 2022). Motifs were then identified using TBtools, and schematic diagrams of conserved motifs in the SNARE proteins were drawn accordingly [68].

### 4.3. Chromosome Location and Gene Duplications in the SNARE Family

The lengths of each chromosome and the position of every *SNARE* gene were obtained from gff3 annotation files in the peanut database, and the chromosomal distribution and density of genes were determined using TBtools. The following two criteria were considered when analyzing *SNARE* gene duplications: (1) the length of the shorter sequence covers at least 70% of the longer sequence; (2) the identity of the two sequences is at least 70%. *SNARE* gene duplications were screened using the MCScanX algorithm with default settings (e ≤ 1 × 10^−10^) [69]. Diagrams were drawn using advanced Circos in TBtools [68].

### 4.4. Promoter Analysis of SNARE Genes in Peanut

Sequences encompassing 2000 bp upstream of the start codon of each *AhSNARE* gene were obtained from the peanut genome using TBtools, and analysis of *cis*-acting elements was performed using the online server PlantCARE [70]. The distribution of motifs in each promoter was drawn using TBtools [68].

### 4.5. RNA-Seq Data and Differential Expression of SNARE Genes

The differential expression of *SNARE* genes in different tissues and at different stages of peanut pod development were analyzed. Specifically, 22 different transcriptome samples were obtained from Genbank BioProject PRJNA291488 [71]. Transcriptome assembly and expression value estimation (Fragments per kilobase million, FPKM) were performed using StringTie (v1.3.4) with default parameters, as reported by Liu et al. [72].

RNA-seq data of roots and leaves of FH1 peanut (‘Fenghua No. 1′) were derived from previous work (Genbank PRJNA553073). Raw sequences were compared to the genome of *A. hypogaea* ‘Tifrunner’ (https://www.peanutbase.org/, accessed on 24 May 2022) using HISAT2 software [73]. Gene expression levels and FPKM values of all genes were obtained using the Featurecounts tool in the Subread software [74]. The expression profiles of *SNARE* genes in different tissues and under diverse stress were extracted and averaged for plotting heatmaps using TBtools.

### 4.6. Plant Materials and Growth Conditions

Peanut seeds (FH1) of uniform size were selected and surface sterilized with 0.1% mercuric chloride; four seeds per bottle were planted in Murashige and Skoog (MS) medium and placed under 16-h light (26 °C)/8-h dark (20 °C) conditions in a culture room for 2 weeks. Seedlings showing the same growth vigor were selected for 8 °C treatment and 200 mM NaCl treatment; each treatment was repeated three times.

### 4.7. RNA Isolation and Expression of SNARE Genes after Stress

After treatment, leaves and roots were separately harvested, frozen immediately in liquid nitrogen and stored at −80 °C. Total RNA was extracted using an FastPure^®^ Plant Total RNA Isolation Kit (Vazyme, Nanjing, China). Ten genes showing significant differential expression after salt and low-temperature treatments were examined using RT-qPCR. Specific primers were designed using Beacon Designer 7.0 software (Appendix A). The housekeeping gene *TUA5* was used as an internal reference with primers, as described by Liu et al. [72]. RT-qPCR was performed using a HiScript III RT SuperMix kit (Vazyme, Nanjing, China), as described by Lu et al. [32]. The 2^−∆∆Ct^ method was used for calculating the relative expression of each *SNARE* gene [36,75].

### 4.8. Functional Identification Using a Yeast Expression System

Full-length cDNA sequences of *AhSYP122a*, *AhSNAP33a* and *AhVAMP721a* were obtained from the genome database of *A. hypogaea*. PCR fragments were amplified from cDNA using specific primers (Appendix A) designed using CE Design software (https://crm.vazyme.com/cetool/simple.html, accessed on 22 July 2022); then, they were purified and cloned into the *EcoR* I sites of pESC-URA (Sangon Biotech, Shanghai, China) using a ClonExpress II One Step Cloning Kit (Vazyme Biotech, Nanjing, China) to generate pESC-*AhSYP122a*, pESC-*AhSNAP33a* and pESC-*AhVAMP721a* plasmids. The *AhSNAP33a* and *AhVAMP721a* coding sequences were inserted at another site of the pESC-*AhSYP122a* vector to generate pESC-*AhSYP122a/AhSNAP33a* and pESC-*AhSYP122a/AhVAMP721a* plasmids, respectively.

The resulting plasmids were introduced into *Saccharomyces cerevisiae* strain BY4741 using the lithium acetate/polyethylene glycol (LiAc/PEG) method. Single colonies of transformant yeasts were picked and inoculated into liquid synthetic defined SD–Ura medium for 2 days at 28 °C (OD_600_ = 0.6). Cultures were then diluted with fresh pre-warmed SD–Ura medium (dilution 1:10) and serially diluted in 10-fold steps; 3-µL aliquots of each dilution were dropped onto SD–Ura medium (2% [*w*/*v*] galactose) plates with or without different concentrations of NaCl for 5 h and at −20 °C for 1 h, respectively. Growth of yeast cells was observed and recorded after 48 h at 29 °C.

## 5. Conclusions

We identified 129 *SNARE* genes in *A. hypogaea*, 63 in *A. duranensis* and 64 in *A. ipaensis*, respectively, which we classified into five groups based on their phylogenetic relationship with their homologs in Arabidopsis. *AhSNARE*s are unevenly scattered on the chromosomes and exhibit a high rate of homoeolog retention. The exon–intron and motif structures of the *SNARE* family are highly conserved. Promoters of *AhSNARE*s contain *cis*-elements associated with development and stress responses. Comprehensive analysis of *AhSNARE* gene expression patterns suggested that *AhVTI13b* plays an important role in the storage of lipid proteins; *AhSYP131b*, *AhSYP72*, *AhNPSN11a/b*, *AhVAMP713* and *AhVAMP723* may play a role in the specific developmental patterns of peanut; and *AhSYP122a*-*AhSNAP33a*-*AhVAMP721a* may play a crucial role in stress responses. This study provides a reference for peanut breeding and further exploration of the mechanism by which SNAREs respond to abiotic stress and development in peanuts.

## Figures and Tables

**Figure 1 ijms-24-07103-f001:**
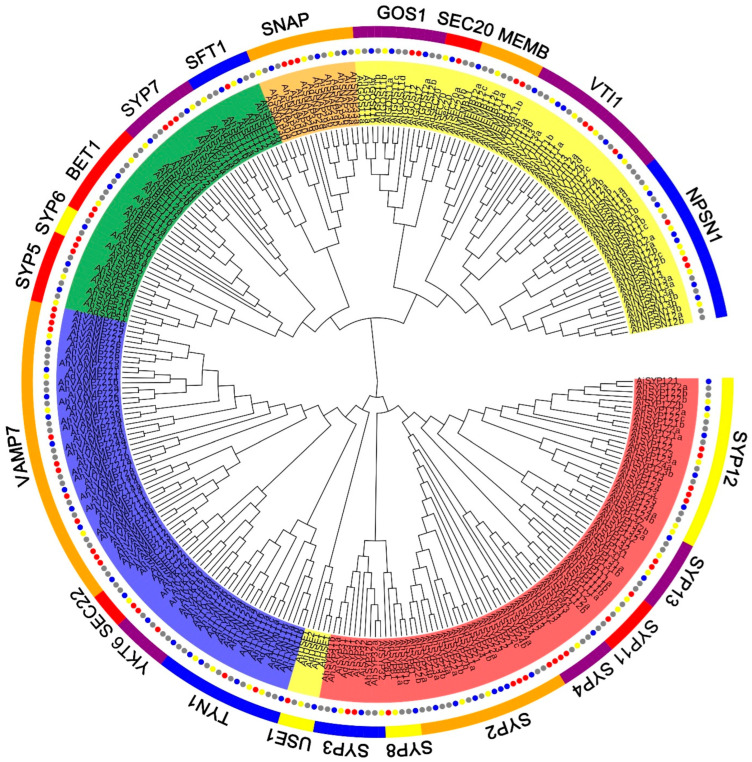
Phylogenetic tree of the SNARE family in *A. hypogaea*, *A. duranensis*, *A. ipaensis* and *Arabidopsis*. Maximum-likelihood tree based on full-length SNARE proteins sequences reconstructed using the nearest-neighbor-interchange model. SNARE protein sequences from *A. hypogaea*, *A. duranensis*, *A. ipaensis* and *Arabidopsis* are marked with gray, blue, green and red circles, respectively. Qa-, Qb-, Qc-, Qb+c and R-SNARE are marked with red, yellow, green, orange and blue backgrounds, respectively. The ribbon in the outer ring indicates different subfamilies.

**Figure 2 ijms-24-07103-f002:**
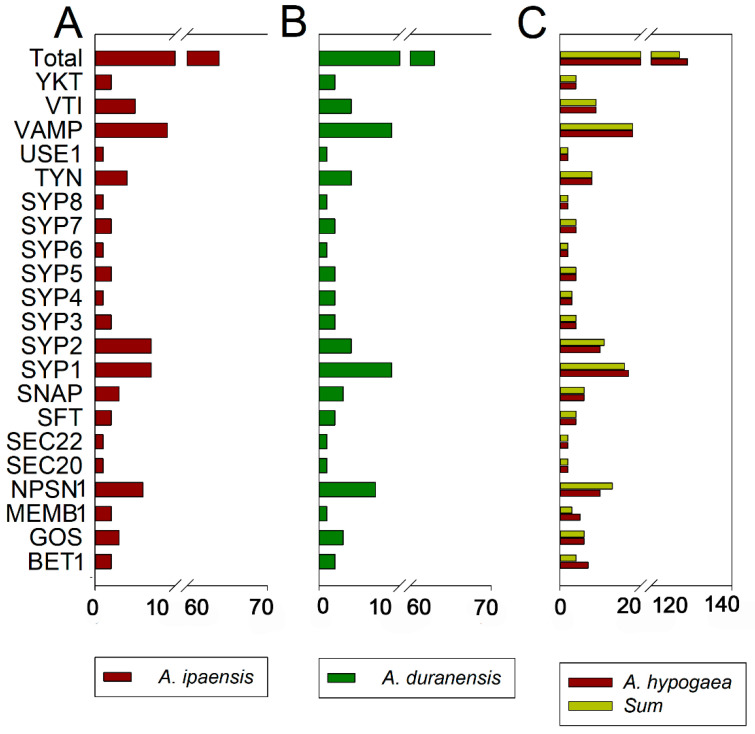
Number of *SNARE* genes identified per SNARE-type subfamily. (**A**) *A. ipaensis*; (**B**) *A. duranensis*; (**C**) *A. hypogaea*. The sum of genes in *A. ipaensis* and *A. duranensis* (green–yellow) is also shown (**C**).

**Figure 3 ijms-24-07103-f003:**
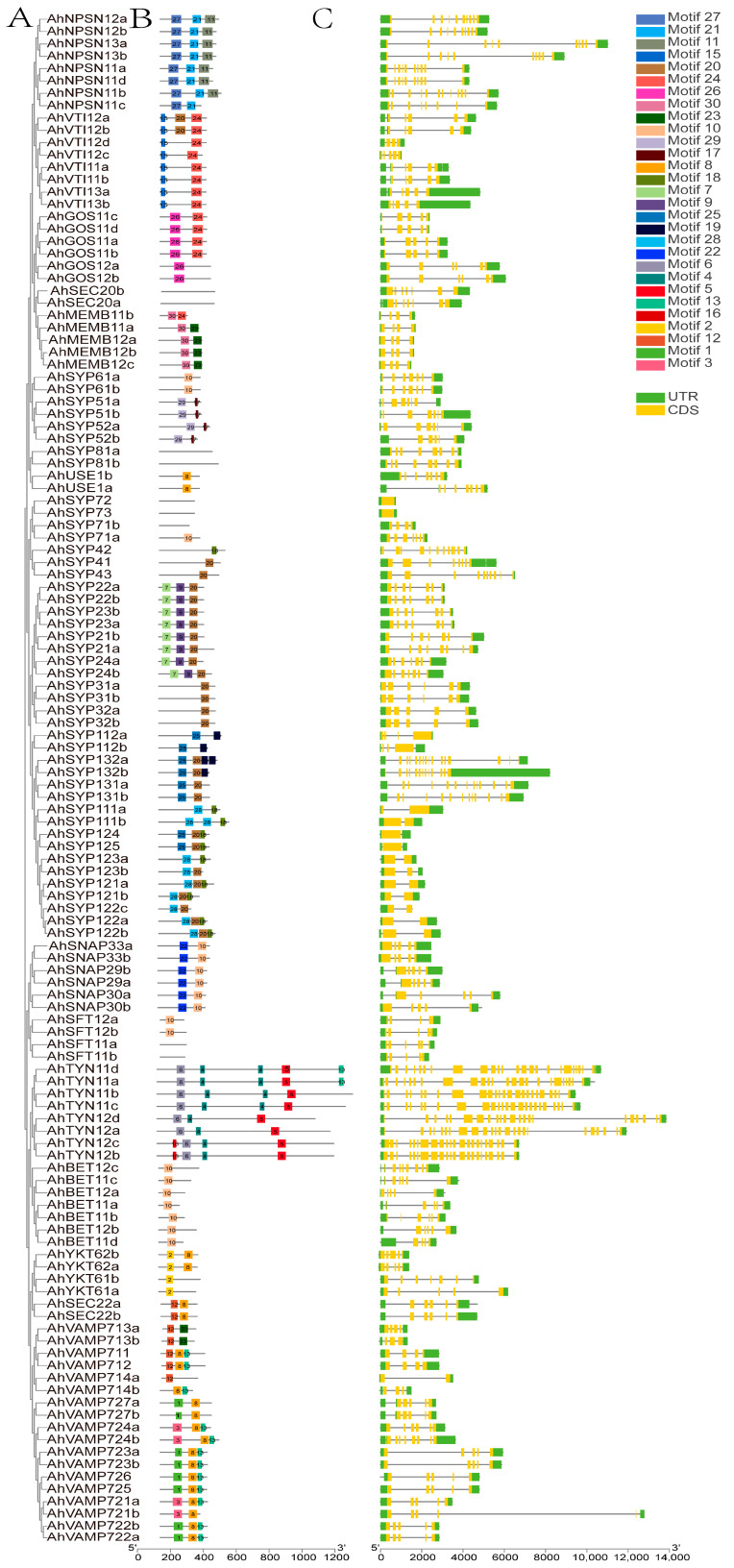
Phylogenetic relationships, gene structures and conserved motifs of AhSNARE genes in cultivated peanut. (**A**) Phylogenetic relationships; (**B**) conversed motifs; (**C**) exon–intron structures. Yellow boxes represent exons; gray lines represent introns.

**Figure 4 ijms-24-07103-f004:**
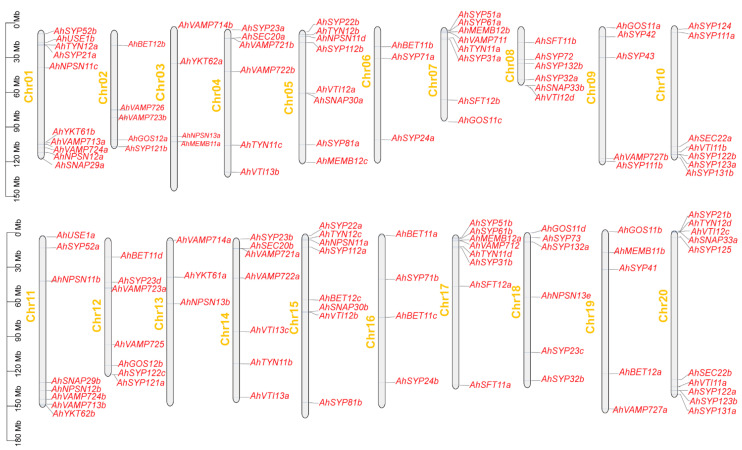
Chromosomal location and distribution of 129 *AhSNARE* genes. Chromosome size is indicated by relative length. Scale bar represents megabases (Mb). Physical locations of *AhSNARE* genes are indicated on each chromosome.

**Figure 5 ijms-24-07103-f005:**
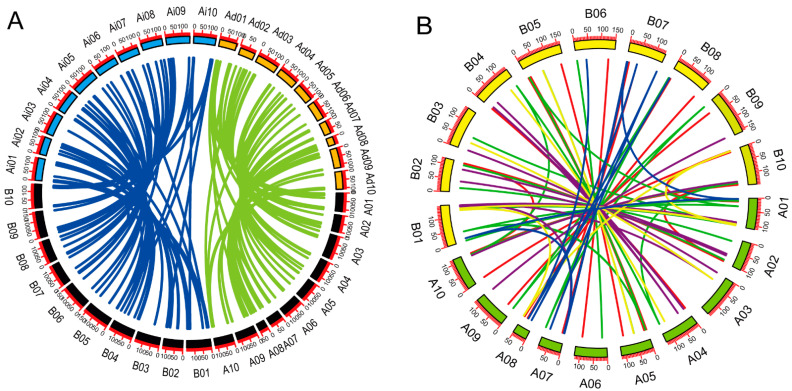
Collinearity analysis of *SNARE* genes. (**A**) Synteny analysis of *SNARE* genes in *A. hypogaea*–*A. duranensis* and *A. hypogaea*–*A. ipaensis*. Green lines indicate syntenic genes between *A. duranensis* and *A. hypogaea*. Blue lines indicate syntenic genes between *A. duranensis* and *A. hypogaea*. Black boxes represent *A. hypogaea* chromosomes. Colored boxes represent chromosomes of *A. duranensis* and *A. ipaensis*. Ad, *A. duranensis*; Ai, *A. ipaensis*; Ah, *A. hypogaea*. (**B**) Synteny analysis of *SNARE* genes within *A. hypogaea*. Colored lines represent syntenic relationships between different type of SNAREs. Red, Qa-SNARE; green, Qb-SNARE; blue, Qb+c-SNARE; purple, R-SNARE. Scale bar represents megabases (Mb). Chromosome numbers are indicated on the top of each bar. Green boxes represent A genome; yellow boxes represent B genome.

**Figure 6 ijms-24-07103-f006:**
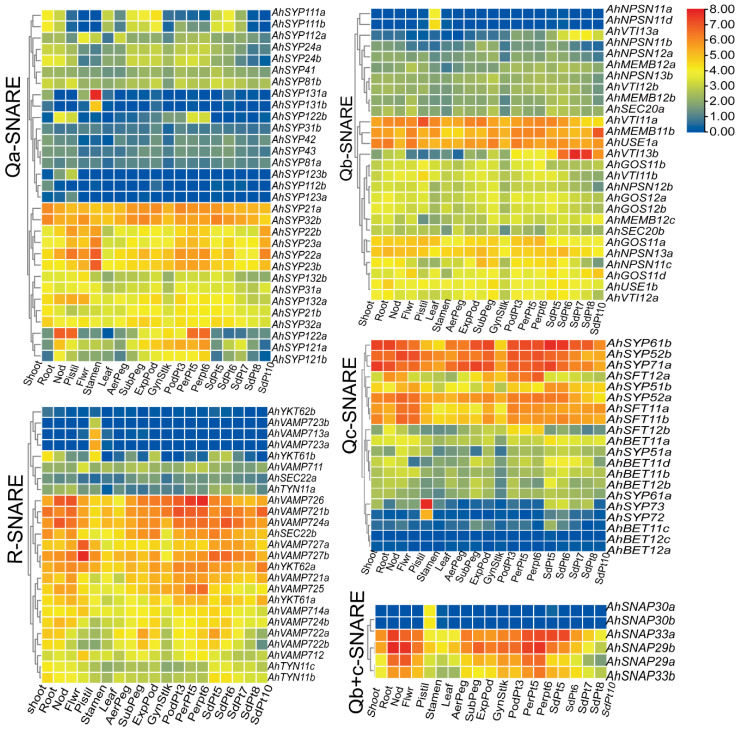
Expression profiles of *SNARE* genes in seven peanut tissues and at 12 different developmental stages of peanut. The heatmap was generated using TBtools software, and fragments per kilobase of transcript per million fragments (FPMK) values for *SNARE* genes were Log2-transformed. The color box represents lower values (blue) to higher values (red).

**Figure 7 ijms-24-07103-f007:**
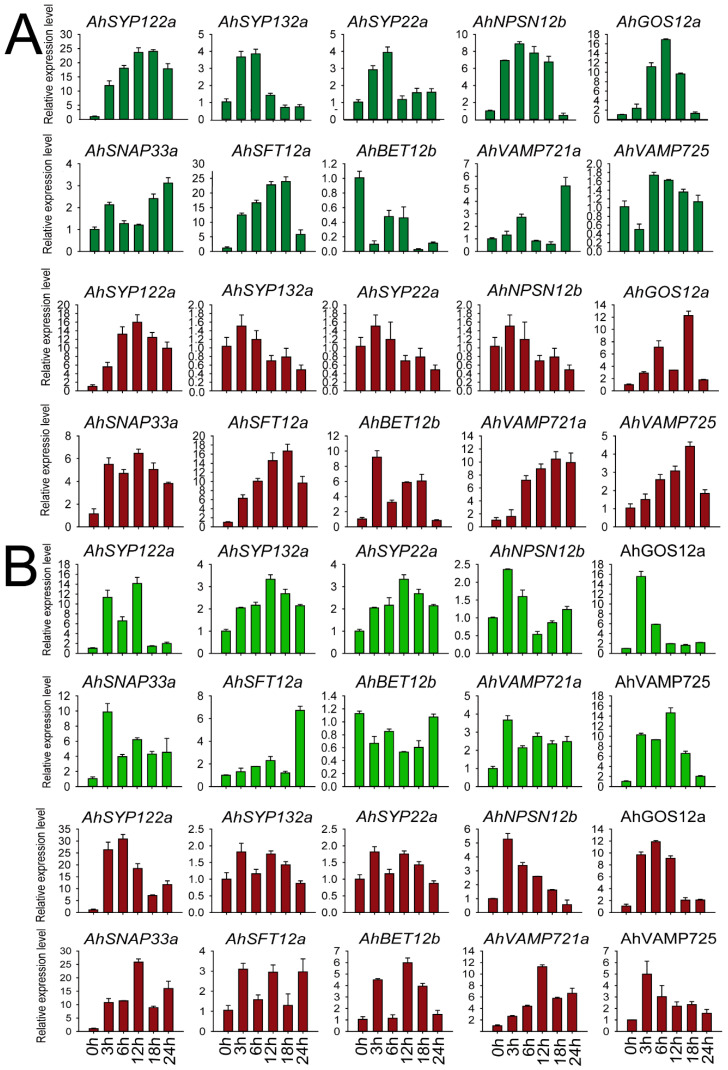
Expression of *AhSNARE* genes in response to salt and chilling stress. Relative expression levels of 10 *AhSNARE* genes in roots (red) and leaves (green) after 0 h, 3 h, 6 h, 12 h and 24 h of salt (**A**) or low-temperature treatment (**B**) as determined by RT-qPCR. Data were calculated from three biological replicates. The error bars show the standard deviation of the three biological replicates.

**Figure 8 ijms-24-07103-f008:**
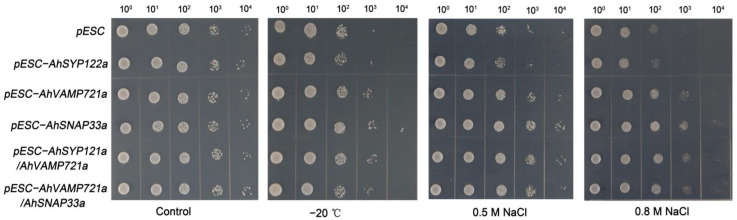
Growth of transformed yeast harboring pESC-*AhSYP122a*, pESC-*AhVAMP721a*, pESC-*AhSNAP33a*, pESC-*AhSYP122a/AhSNAP33a* or pESC-*AhVAMP721a/AhSNAP33a* vectors under −20 °C for 1 h, and 0.5 M or 0.8 M NaCl for 5 h. The experiment was repeated three times.

## Data Availability

The data and materials that were analyzed in the current study are available from the corresponding author upon reasonable request.

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
