# Peer review of "Genome-Wide Analysis of the SNARE Family in Cultivated Peanut (Arachis hypogaea L.) Reveals That Some Members Are Involved in Stress Responses"

_ijms, 2023, doi:10.3390/ijms24087103_

Round 1

Reviewer 1 Report

In this manuscript the identification of SNARE genes, proteins that mediate membrane fusion during vesicular transport between endosomes and the plasma membrane, was carried out in peanut plants. A total of 129 genes were identified in the genomes of peanut and using bioinformatics the structure and chromosomal location was presented.

In addition, gene expression was analysed at different developmental stage of the plant and tissue as well as under salt and chilling stress.

The manuscript is well written in English and the methodology is correct.

However, there is a poor discussion since a physiological point of view of the role of SNARE proteins in each specific tissue (root, leaves as example) during each developmental stage, what type of proteins can be moved by trafficking through the plasma membrane during these developmental stages and which are their functions.

Similarly, the role of SNARE under chilling and salt stress is not well described since a physiological point of view. Which implications have that during salt stress or chilling, SNARE may allow the migration to the plasma membrane of proteins as potassium transporters or aquaporins?

Also, discussion about the role of conserve motif in SNARE can be considered and how differences in the sequence (isoforms) may condition or not different SNARE functions?

Deeper discussion considering these aspects should improve the manuscript.

Author Response

Thank you very much for your critical reading and revision of the manuscript (ijms-2278307; title: Genome-wide analysis of the SNARE family in cultivated peanut (Arachis hypogaea L.) reveals that some members are involved in stress responses).   The following is the detailed explanation how we complied with the reviewers’ suggestions. We hope that you or the reviewers will point out the errors that we can correct during further revision.

Comments from the Editors and Reviewers:

In this manuscript the identification of SNARE genes, proteins that mediate membrane fusion during vesicular transport between endosomes and the plasma membrane, was carried out in peanut plants. A total of 129 genes were identified in the genomes of peanut and using bioinformatics the structure and chromosomal location was presented.

In addition, gene expression was analysed at different developmental stage of the plant and tissue as well as under salt and chilling stress.

The manuscript is well written in English and the methodology is correct.

However, there is a poor discussion since a physiological point of view of the role of SNARE proteins in each specific tissue (root, leaves as example) during each developmental stage, what type of proteins can be moved by trafficking through the plasma membrane during these developmental stages and which are their functions.

Similarly, the role of SNARE under chilling and salt stress is not well described since a physiological point of view. Which implications have that during salt stress or chilling, SNARE may allow the migration to the plasma membrane of proteins as potassium transporters or aquaporins?

Also, discussion about the role of conserve motif in SNARE can be considered and how differences in the sequence (isoforms) may condition or not different SNARE functions?

Deeper discussion considering these aspects should improve the manuscript.

Response: Thanks for the suggestions, we have accepted your advice and deeper discussion has been added. Several critical periods of peanut fruit development were discussed in line 473-478, 482-484, 489-497. The role of vesicular transport under abiotic stress was further discussed in line 519-520 and marked in red in the manuscript.

Reviewer 2 Report

 In this paper the authors describe SNARE gene family in cultivated pea-nut and its ancestors,and found that AhVTI13b plays an important role in the storage of lipid proteins, AhSYP122a, AhSNAP33a and AhVAMP721 are involved in cold and salt stress.  I consider that the work is well done, because not too much research on SNARE genes has been carried out on this plant. However, there are a number of errors, some of which are important and must be corrected by the authors. The introduction is very well planned and written.

1. The authors must follow the journal rules (in the introduction section delete the author names and add No instead)

2. L23, L25  The name of all genes must always appear in italics.

L13,16,26  The Latin name of plant must be in italics

3. L376 “H2O2” typo

4.  L571  revise all the references and follow the rules.

5. L483 play key roles in cell-plate formation and their encoding genes are

specifically expressed during cell division Reference needed.

6. Fig 8  The gene names in Figure 8 need to be re-edited and should be consistent with those expressed in the article.

Author Response

Thank you very much for your critical reading and revision of the manuscript (ijms-2278307; title: Genome-wide analysis of the SNARE family in cultivated peanut (Arachis hypogaea L.) reveals that some members are involved in stress responses).   The following is the detailed explanation how we complied with the reviewers’ suggestions. We hope that you or the reviewers will point out the errors that we can correct during further revision.

Comments from the Editors and Reviewers:

 In this paper the authors describe SNARE gene family in cultivated pea-nut and its ancestors,and found that AhVTI13b plays an important role in the storage of lipid proteins, AhSYP122aAhSNAP33a and AhVAMP721 are involved in cold and salt stress.  I consider that the work is well done, because not too much research on SNARE genes has been carried out on this plant. However, there are a number of errors, some of which are important and must be corrected by the authors. The introduction is very well planned and written.

  1. The authors must follow the journal rules (in the introduction section delete the author names and add No instead)

Response: Thanks for the suggestions, we have accepted your advice and re-edited the references and replaced the author's name with No.. 

  1. L23, L25  The name of all genes must always appear in italics.

L13,16,26  The Latin name of plant must be in italics

Response: Done in line 13, 16, 18, 23, 24, 25, 26 

  1. L376 “H2O2” typo

Response: Done in line 370.

  1. L571  revise all the references and follow the rules.

Response:  Thanks for the suggestions, we have accepted your advice and re-edited the references. 

  1. L483 “play key roles in cell-plate formation and their encoding genes are

specifically expressed during cell division” Reference needed.

Response: Thanks for the suggestions,we have add related references in line 488 in the manuscript.

  1. Fig 8  The gene names in Figure 8 need to be re-edited and should be consistent with those expressed in the article.

Response: Thanks for the suggestions,We have re-edited the figures in the article.